# Cardiological Reference Intervals in Adult American Staffordshire Terrier Dogs

**DOI:** 10.3390/ani13152436

**Published:** 2023-07-27

**Authors:** Oktawia Szpinda, Marta Parzeniecka-Jaworska, Michał Czopowicz, Izabella Jońska, Joanna Bonecka, Michał Jank

**Affiliations:** 1Department of Pathology and Veterinary Diagnostics, Institute of Veterinary Medicine, Warsaw University of Life Sciences-SGGW, Nowoursynowska 159c, 02-776 Warsaw, Poland; marta_parzeniecka-jaworska@sggw.edu.pl; 2Division of Veterinary Epidemiology and Economics, Institute of Veterinary Medicine, Warsaw University of Life Sciences-SGGW, Nowoursynowska 159c, 02-776 Warsaw, Poland; michal_czopowicz@sggw.edu.pl; 3Department of Small Animal Diseases with Clinic, Institute of Veterinary Medicine, Warsaw University of Life Sciences-SGGW, Nowoursynowska 159c, 02-776 Warsaw, Poland; izabella_jonska@sggw.edu.pl (I.J.); joanna_bonecka@sggw.edu.pl (J.B.); 4Division of Pharmacology and Toxicology, Department of Preclinical Sciences, Institute of Veterinary Medicine, Warsaw University of Life Sciences-SGGW, Ciszewskiego 8, 02-787 Warsaw, Poland; michal_jank@sggw.edu.pl

**Keywords:** echocardiography, electrocardiography, reference values, vertebral heart score, VHS

## Abstract

**Simple Summary:**

The American Staffordshire Terrier (AST) is a canine breed characterized by a very specific body build, high activity, and exuberant temperament. The predispositions of this breed to particular cardiovascular diseases are not known. Breed-specific reference intervals (RI) for echocardiography have been recently established for AST dogs, however, no breed-specific RI for the electrocardiography (ECG), vertebral heart size (VHS), and arterial systolic blood pressure are available. Our study indicates that dogs of the AST breed should be individually approached and the condition of their heart should be assessed against breed-specific RI. Such action will increase the chance to detect even minor deviations from the breed-specific RI and prevent misdiagnosis.

**Abstract:**

The aim of this study was to determine reference intervals (RI) for echocardiography, electrocardiography (ECG), vertebral heart score (VHS) measurement, and arterial systolic blood pressure (SBP) in American Staffordshire Terrier dogs. The study population included 29 clinically healthy AST dogs of different ages, genders, and body weights. SBP measurement, ECG, thoracic radiography, and echocardiography were performed on each dog. Compared to RIs available for the general population of dogs, the duration of the P wave and QRS complex was longer and VHS was higher. Moreover, the left ventricular dimension in diastole and systole, left atrial dimension, and end point to septal separation values were higher, while the interventricular septum in diastole and systole and aortic root diameter were lower compared to general similar average body weight. The AST breed has a different heart shape, which in this breed is more rounded compared to other dog breeds, especially the deep chest. The specific body structure and the shape of the heart had an impact on the results of the cardiological examination.

## 1. Introduction

Domestic dogs (*Canis familiaris*) are the most diverse mammal species in the world. Body weight (BW) can range from <1 kg to >100 kg [1]. Such a morphological heterogeneity hampers developing a single reference interval (RI) for any imaging diagnostic method. This fact becomes gradually acknowledged in the veterinary practice by developing RIs adjusted by BW or body surface area (BSA), the latter being directly derived from BW. However, even dogs of the same BW may have completely different body shapes and structures that might modify the three-dimensional appearance of examined organs and negatively affect the accuracy of diagnostic methods. This variability is, to some extent, associated with the genotype of the breed to which a dog belongs. An increasing knowledge of between-breed differences has stimulated the development of breed-specific RIs for various diagnostic tests used in canine cardiology, including thoracic radiography (TXR) [2,3,4], electrocardiography (ECG) [5,6], and echocardiography [7,8,9,10].

The American Staffordshire Terrier (AST) is one of the canine breeds known for its distinctive body conformation. These dogs are distinguished by a barrel chest, compact back, and very well-developed musculature. They have exceptional strength in relation to their BW and height. A clearly marked sexual dimorphism is also observed—males are taller, heavier, and much more muscular compared to females [11]. BW in adult females and males is 18–25 kg and 25–32 kg, respectively, while an ideal height at the withers is 43–46 cm and 46–48 cm, respectively [1]. In recent years, the AST has gained great popularity all over the world.

However, not much is known about the AST predilection for heart diseases. Together with bulldogs and standard schnauzers, they have been shown to be strongly predisposed to congenital pulmonic stenosis [12,13]. Analogically to other large canine breeds, they are believed to be commonly affected by dilated cardiomyopathy (DCM), although clear epidemiological evidence is lacking. An accurate diagnostics of heart diseases in AST requires the application of well-developed breed-specific RIs. Such RIs for echocardiographic measurements developed in 57 adult ASTs have only been recently published [14], whereas they do not exist for TXR and ECG. Without such RI, the determination of AST predilection to any kind of heart disease is almost impossible. Therefore, we carried out this study to determine RIs for echocardiography, ECG, vertebral heart score (VHS) measurement, and arterial systolic blood pressure (SBP) in ASTs.

## 2. Materials and Methods

### 2.1. Study Population

The study was carried out at the Small Animal Clinic of the Warsaw University of Life Sciences-SGGW on client-owned patients of the clinic. All examinations were performed by one person (OS). Adult dogs of the AST breed confirmed by a formal document issued by the Polish Association of Pedigree Dog Breeders were enrolled in the study. The dogs’ owners were informed about the purpose and protocol of the study, and they all granted informed written consent for participation in the study. In each dog, the following procedures were performed: standard clinical examination, SBP measurement, hematological and biochemical blood check-up, ECG, echocardiography, and TXR. Only measurements obtained from dogs with normal results of clinical examination and blood check-ups were used for the determination of the RIs.

### 2.2. Evaluation of General Health Status

The standard clinical examination in a standing position was performed in a quiet room. It included measuring the rectal body temperature, a visual evaluation of mucus membranes of the eyelids and gums measuring the capillary refill time, palpation of superficial lymph nodes (submandibular, prescapular, and popliteal), palpation of the abdomen, and auscultation of the heart and lungs.

SBP was measured with Doppler ultrasonic sphygmomanometry using a VetBP Doppler Kit (VetBP, Poland). The cuffs were individually selected according to the dog’s size. The width of the cuff was 30–40% of the circumference of the extremity and tail at the site of cuff placement. The examination was preceded by at least a 10-min acclimatization in the consultation room. SBP was measured 5 times, the lowest and highest values were discarded, and the arithmetic mean of the middle 3 measurements was considered the final SBP.

Blood was collected from the lateral saphenous vein to the tube with EDTA anticoagulant for routine hematology with manual smear and to the dry tube for routine biochemistry. Blood check-up results were interpreted according to the RIs of the laboratory [15].

### 2.3. Cardiological Examination

An ECG was performed using a BTL-08-MD device (BTL, Warsaw, Poland). A 6-lead limb record (leads I, II, III, AvR, aVL, aVF) was recorded. The following measurements obtained in the lead II were analyzed: the amplitude [mV] and duration [s] of the P wave, the duration [s] of the PQ interval, the amplitude of the Q, R, and S waves [mV], the duration of the QRS complex [s], the duration of the QT interval [s], the ST segment elevation/lowering, the amplitude of the T wave [mV], and the mean electrical axis (MEA) of the heart. The heart rate (HR) was also measured [beats/min] and the cardiac rhythm was qualitatively assessed. All ECG measurements were performed according to standard recommendations [16]. MEA was calculated in the online calculator (Mean Electrical Axis Calculator from vin.com portal).

The transthoracic echocardiography was performed using an Aloka 4000 ultrasound machine (Miro, Warsaw, Poland) with a 3.5 MHz sector transducer and an Esaote MyLab X8 machine with a 3.5 MHz sector transducer (Esaote, Warsaw, Poland) with simultaneous ECG monitoring using electrodes connected to the skin folds. During the project, the ultrasound machine was replaced with a new one; therefore, the tests were carried out on two different machines. The dogs were examined in a standing position due to their temperament. In the standing position, they were calm, and when laid down, they squealed, trembled, and panted, which made it impossible to perform a precise echocardiographic examination. The hair was not clipped; only an appropriate amount of alcohol and coupling gel was used to ensure tight contact between the transducer and skin.

The heart was assessed in four views. In the right parasternal short-axis view and two-dimensional (2D mode) presentation, the diameter of the left atrium (LA) and aortic root (Ao) were measured. Moreover, the systolic blood flow velocity through the pulmonary artery [m/s] (right ventricular outflow tract, RVOT) was measured [m/s] using Doppler ultrasonography. The ratio of the left atrium to the aorta (LA/Ao) and the right ventricular outflow tract obstruction [mmHg] were automatically calculated.

In the right parasternal short-axis papillary muscle view at the height of the papillary muscles, below the mitral valve annulus in a one-dimensional (M-mode) presentation, the following measurements were taken [cm]: the interventricular septum thicknesses in end-diastole (IVSd) and end-systole (IVSs), left ventricular peripheral (free) wall in end-diastole (LVWd) and end-systole (LVWs), end-diastolic right ventricular dimension (RV) and left ventricular dimension in end-diastole (LVDd) and end-systole (LVDs) [17]. The LVDs and LVDd were indexed to BW according to Cornell’s method [8] to obtain their normalized measurements (LVDdN and LVDsN, respectively). At the mitral height, the distance of the septal mitral valve (point E) from the interventricular septum (EPSS) was measured [cm].

The left ventricular fractional shortening (FS) was calculated as follows:FS = (LVDd − LVDs)/LVDd × 100%

The end-diastolic and end-systolic left ventricular volume (EDV and ESV, respectively) were measured in the 2D presentation right parasternal long-axis view (PLAX) and left apical four-chamber view (A4C) using Simpson’s method of discs (SMOD) [ml]. In the A4C view, mitral valve E wave velocity and mitral valve A wave velocity were also measured [m/s] and their ratio (E/A) was calculated. EDV and ESV were divided by body surface area (BSA) to obtain EDV and ESV indices in mL/m^2^ (EDVI and ESVI, respectively) [18]. BSA was calculated as follows [19]:BSA [m^2^] = 0.101 × BW [kg] ^2/3^

The ejection fraction (EF) was calculated as follows:EF = (EDV − ESV)/EDV × 100%

In the left apical five-chamber view and two-dimensional presentation, the velocity of blood flow through the aorta (LVOT) was measured [m/s] and the left ventricular outflow tract obstruction [mmHg] was automatically calculated.

Thoracic radiography was performed in the right lateral and sagittal (dorsoventral, DV) projection and the vertebral heart score/size (VHS) was calculated according to the method of Buchanan and Bucheler [3]. As radiographs were carried out in non-sedated dogs, sagittal views had low quality, which precluded a reliable calculation of VHS. Therefore, only VHS from lateral views was analyzed.

### 2.4. Statistical Analysis

Numerical variables were presented as the arithmetic mean and standard deviation (SD) if normally distributed; otherwise, they were presented as the median and interquartile range (IQR). The normality of distribution was evaluated based on histograms, quantile-quantile (Q-Q) plots, and using the Shapiro-Wilk test. The asymmetry of distribution was assessed using Pearson’s coefficient of skewness (CoS, with 95% confidence intervals, CI 95%) and classified as significantly right-hand skewed (if the entire CI 95% of CoS was above 0) or significantly left-hand asymmetric (if the entire CI 95% of CoS was below 0) or symmetric (if CI 95% of CoS included 0). The shape of the distribution (heaviness of the tails of a distribution) was assessed using the coefficient of concentration (kurtosis, with the CI 95%) and classified as significantly leptokurtic (i.e., of thin tails, if the entire CI 95% of the kurtosis coefficient was above 0) or significantly platykurtic (i.e., of heavy tails, if the entire CI 95% of the kurtosis coefficient was below 0) or normocurtic (if the CI 95% of the kurtosis coefficient included 0) [20,21]. The outliers were identified using Horn’s algorithm based on Tukey’s interquartile fence. Cardiological measurements (VHS, SBP, ECG, and echocardiographic) were compared between males and females using the unpaired Student’s *t*-test if normally distributed; otherwise, the Mann-Whitney U test was used. The relationship between the cardiological measurements and age or body weight was investigated using Pearson’s product-moment correlation coefficient (R) if normally distributed; otherwise, Spearman’s rank correlation coefficient (R_s_) was used. The strength of correlation was classed as follows [22]: R or R_s_ = 0.00 to 0.19—very weak, 0.20 to 0.49—weak, 0.50 to 0.69—moderate, 0.70 to 0.89—strong, and 0.90 to 1.00—very strong. All statistical tests were two-tailed and a significance level (α) was set at 0.05. Statistical analyses were performed using TIBCO Statistica 13.3 (TIBCO Software Inc., Palo Alto, CA, USA).

Reference intervals (RIs) for cardiological measurements were estimated using the standard parametric method on either untransformed data (untransformed standard method, UTS) if normally distributed or Box-Cox transformed data (Box-Cox transformed standard method, BCTS) in the case of a non-normal distribution of data. RIs were presented as the lower and upper reference limits along with their 90% confidence intervals (CI 90%) calculated using the bootstrap method [23]. The calculative methods used were based on the guidelines of the American Society for Veterinary Clinical Pathology (ASVCP) [24] and RIs were calculated in the Reference Value Advisor version 2.1 [25].

## 3. Results

### 3.1. Study Population and Clinical Condition

As part of the project, 91 AST dogs were tested, of which 29 clinically healthy dogs with blood check-ups within RIs were qualified for the final research group. There were 15 males and 14 females, aged from 1.5 to 10 years with a median (IQR) of 4.5 (2.5–5.8) years. Their BW ranged from 23 to 44 kg with a median (IQR) of 30 (27–32) kg and their BCS varied from 3 to 8 with a median of 5. Males did not significantly differ from females with respect to age (median (IQR) of 3.9 (2.0–5.4) vs. 4.8 (3.4–6.6) years, *p* = 0.274), BW (median (IQR) of 30 (26–35) vs. 30 (27–30) kg, *p* = 0.287), or BCS (median of 5 in both genders, *p* = 0.975).

None of the dogs had cardiac murmurs or pathological arrhythmias. Eleven dogs (37.9%) had respiratory sinus arrhythmia in auscultation. SBP ranged from 80–180 mmHg with an arithmetic mean (±SD) of 133 ± 23 mmHg (Appendix A). SBP was not significantly linked with gender (*p* = 0.407), age (*p* = 0.998), or BW (*p* = 0.504) (Appendix A). The RI was very wide—from 85 to 180 mmHg (Appendix A).

### 3.2. ECG

All dogs had a sinus rhythm without any pathological arrhythmias. Respiratory sinus arrhythmia was observed in eight dogs (27.6%). Heart rate ranged from 70 to 150 beats/min with an arithmetic mean (±SD) of 109 ± 19 beats/min. QRS complexes in lead II had a qRs morphology. In twenty-six dogs (89.6%), the ST segment was reduced with respect to the isoelectric line; in two dogs (6.9%), it was elevated, and in one (3.5%) dog, it was consistent with the isoelectric line. In all dogs, the T wave was single-phase with positive polarity. MEA was shifted to the left (left-gram) in 17 dogs (58.6%); the remaining 12 dogs had nomograms. The QRS duration was significantly longer in females (*p* = 0.002). The amplitudes of the R wave (*p* = 0.014) and T wave (*p* = 0.039) were significantly higher in males. The Q amplitude was significantly negatively correlated with age (R = −0.47; *p* = 0.010) and the QT duration was significantly negatively correlated with body weight (R_s_ = −0.37; *p* = 0.046) (Appendix A). Both correlations were weak.

Compared to commonly accepted RIs, the duration of the P wave in 18 AST dogs (62.1%) exceeded the commonly accepted reference value of 0.04 s, and in 19 AST dogs (65.5%), the duration of the QRS complex was longer than the accepted reference value of 0.06 s (Appendix A).

### 3.3. Echocardiography

A significant positive correlation with age was observed for LVDs (R = 0.40; *p* = 0.033), ESV PLAX (R = 0.38; *p* = 0.041), LVDsN (R = 0.37; *p* = 0.049), and ESVI PLAX (R = 0.39; *p* = 0.038). On the other hand, LA (R = −0.39; *p* = 0.037) and RVOT (R = −0.61; *p* = 0.001) were significantly negatively correlated with age. Only the latter correlation was moderate in strength—the flow through the pulmonary valve was slower in older dogs (Figure 1). The others were only weak (Appendix A).

Four measurements were significantly positively correlated with body weight: IVSd (R = 0.49; *p* = 0.007), LVDd (R = 0.55; *p* = 0.002), EDV PLAX (R = 0.40; *p* = 0.033), and ESV PLAX (R = 0.41; *p* = 0.028). The former two correlations were moderate in strength—heavier dogs had a bigger left ventricle (Figure 2A) and a thicker interventricular septum (Figure 2B) (Appendix A).

AST dogs had larger LVDd, LVDs, LA (↑2.59 cm), and EPSS values compared to RIs taken from the general population of dogs of similar average body weight (30 kg). These dogs, compared to the general population, also had thinner IVS in both diastole and systole, as well as smaller Ao diameters. In two dogs (6.9%), LVOT was accelerated and it was 2.2 m/s in both cases (Appendix A).

### 3.4. Thoracic Radiography

The VHS ranged from 9.9–12.2 thoracic vertebral length with an arithmetic mean (±SD) of 10.9 ± 0.6 (Appendix A). It did not significantly differ between males and females, neither was it significantly correlated with age or BW (Appendix A). Compared to commonly accepted RIs (8.5–10.6), AST dogs had higher VHS.

## 4. Discussion

Our study provides breed-specific RIs of various cardiological diagnostic methods. They apply to both genders of adult ASTs. We determined RIs using methods recommended in commonly accepted guidelines of the ASVCP [24]. Even though the guidelines were primarily developed for veterinary clinical pathology, since their publication in the early 2010s, they have been adopted in clinical veterinary medicine and used in many studies on RIs in veterinary cardiology in dogs [14,26,27,28,29,30,31,32,33,34,35,36,37,38,39], as well as other animal species [40]. In most of the aforementioned studies [14,27,29,30,31,32,33,35,36,37,40], calculations were performed using the open-access software— Reference Value Advisor version 2.1. This is most likely because this software offers simultaneous analyses based on several different calculative methods and assists in choosing the most appropriate one. Thus far, in the studies carried out, three analytical approaches have been used—the non-parametric method with CI 90% for reference limits determined using the bootstrap method [14,32,33,36,37,38], the robust method [28,29,31,39], and the standard parametric method based on untransformed or Box-Cox transformed data [26,35]. Very few studies could use the classical non-parametric approach [27,34], since at least 120 dogs have to be enrolled [23]. According to the ASVCP guidelines, we applied the standard parametric method because our data either had or could easily be transformed into Gaussian distribution. Unfortunately, our study was based on a relatively small number of AST dogs—only 29. Although this was still above the lowest accepted number according to the ASVCP guidelines (i.e., 20), the RI studies carried out thus far have enrolled between 41 and 134 dogs of a specific breed, except for one very large study comprising 1331 dogs of various breeds [34]. The small sample size is the main limitation of our study. It resulted from the fact that we applied very stringent criteria for dog selection. Among other criteria, the dogs had to be free from any cardiac murmurs and have perfect blood check-up results. This forced us to drop many of the initially examined dogs and resulted in a relatively small sample of perfectly healthy adult AST dogs. To mitigate the negative impact of the small sample size on the study results, we very carefully evaluated the shape of the distribution of each individual cardiological measurement so that we could choose the most appropriate calculative method. Nevertheless, the obtained RIs should still be applied to clinical practice with the highest caution. Moreover, the small sample size considerably reduced the power of all statistical analyses and between-group comparisons. As a consequence, the potentially insignificant results may only be so due to the low number of compared animals (e.g., males and females), and therefore, should not be considered as a source of any ultimate conclusions.

SBP in the 27 examined dogs, despite their exuberant temperament, was within the generally accepted RI, below 150 mmHg [41]. In two dogs, arterial pressure was beyond the RI (mean systolic arterial pressure of 167 mmHg and 178 mmHg). In these dogs, additional tests, such as a urine test and an ophthalmological examination, did not confirm hypertension. In accordance with the ACVIM guidelines from 2018, a control blood pressure measurement was also recommended after 2–4 weeks [42]. There was a considerable discrepancy between blood pressure values in individual dogs. None of the tested dogs showed signs of hypertension or had any diseases that could cause hypertension, therefore, it can be assumed that all dogs had normal SBP. The range of results obtained could be caused by the very strong emotional arousal of the breed. A one-time pressure finding above 160 mmHg is not any basis for diagnosing hypertension. In the case of overactive patients, it is worth waiting at least 10 min in calm conditions before taking measurements; however, in the case of ASTs, this rule did not bring the desired effect and waiting did not sufficiently calm the dogs. Blood pressure measurements were much easier to take on the tail, as the dogs were hypersensitive when their paws were touched. In the AST breed, in order to better assess blood pressure, it should be measured several times at intervals and then the obtained results should be compared. Hypertension should not be diagnosed based on only one high measurement.

In the electrocardiographic study, in most of the AST dogs, the durations of the P wave and QRS complex were longer than the accepted RI for the general population of dogs. Additionally, in more than half of the dogs, the electrical axis of the heart was shifted to the left. These results may indicate signs of enlargement of the left atrium and left ventricle of the heart, which in turn was not confirmed by echocardiography. The obtained electrocardiographic results in the AST dogs are due to a different heart shape, which in this breed is more rounded compared to dog breeds with deep chests (e.g., Doberman Pincher) [43]. A rounded shape of the heart was also observed in echocardiographic and radiological examinations. A wide range of HR was observed, which was caused by the temperament of the breed.

The echocardiographic examination performed on the AST dogs showed that this breed is characterized by a symmetrical enlargement of the cardiac silhouette compared to generally accepted RIs. AST dogs had larger LVDd, LVDs, and LA, which were similar to those measured in larger dog breeds of 30–40 kg, such as the Doberman Pincher, with a ranging BW of 26–53 kg [44], as well as Boxers (BW 19–41 kg) [45], and German Shepherds (BW 22–37 kg) [46]. The EPSS value was elevated (in most cases, close to the upper limit of the adopted standard), which indicates a rounded heart shape. AST dogs also had smaller Ao diameters; similar sizes of the Ao were examined in Labrador Retrievers [47] and Golden Retrievers [48]. The examined dogs had thinner IVSd and IVSs, similar to the Border Collie breed, with an average body weight of 19 kg [49]. AST dogs had a higher LVOT than the general dog population, similar to that measured in Boxers [45].

The echocardiographic measurements obtained from a Polish population of ASTs are comparable to the RI for the AST breed published in 2021 [14]. Dogs examined in our study had a higher BW than the population examined by Vezzosi’s team, which explains the slightly higher measurements of LVDs (Appendix A). Compared to the aforementioned reference values, our AST study population also had slightly lower LVOT and RVOT (Appendix A). Additionally, no dog had a heart functional murmur in the present study, which was recorded in the dog population studied by Vezzosi’s team.

Due to the great variety of dog breeds subjected to echocardiography examination, RIs for echocardiographic examination for individual dog breeds have been increasingly published in the literature. The authors of these publications have indicated that creating standards for individual breeds based on specific studies of entire populations of dogs of one breed is much more accurate, because it takes into account their breed characteristics—body structure, conformation, and natural physical activity, not only BW. A good example are English bull terriers—a breed similar in body structure and character to AST [50], the Ao of which is smaller than the RI adopted in dogs weighing 18–30 kg. This parameter was comparable to the standards adopted for dogs weighing 8–19 kg; e.g., Welsh Pembroke Corgi [48]. Bullterriers, like AST dogs, had larger LA. In addition, the FS% was lower than in dogs of a similar size and comparable to giant breeds [50].

The mean VHS value in AST dogs was 10.6, which is higher than the reference value established for the general population of dogs. It is known that the VHS value specified for the general population of dogs is only moderately applicable to some breeds [51]. Studies have shown that VHS measurements outside of the breed are also influenced by BW and body structure, gender, and physical activity, as well as projection in which thoracic radiography is performed [2,4,52,53]. In dogs of the AST breed, as in other examined barrel-chested breeds, the VHS value is higher than the generally accepted RI (e.g., Pug, Boston Terrier, Yorkshire Terrier) [4]. The AST dogs, due to their temperament, caused great difficulty during radiographic examinations, which precluded obtaining proper, motionless projections.

The limitation of our study was the method by which the blood pressure in AST dogs was measured. According to the current ACVIM guidelines from 2018 [42], the first measurement should be discarded, and the subsequent five to seven values should be recorded. In most of the examined AST dogs, due to their exuberant temperament, the first and last blood pressure values differed from the others. In addition, an increase in heart rate was observed at the beginning and at the end of the blood pressure test, which was associated with the impatience and emotions of the tested dogs. Therefore, the authors decided to use the old method, which, in their opinion, was more suitable for this breed of dog.

## 5. Conclusions

Our study indicates that dogs of the AST breed should be individually approached and the condition of their heart should be assessed against breed-specific RI. Such action will increase the chance to detect even minor deviations from the breed-specific RI and prevent misdiagnosis.

## Figures and Tables

**Figure 1 animals-13-02436-f001:**
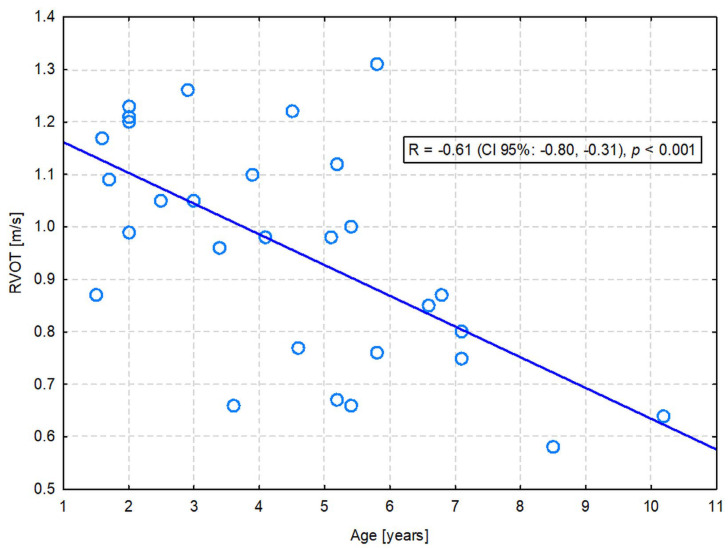
The moderate negative correlation between the right ventricular outflow tract (RVOT) velocity and the age of dogs.

**Figure 2 animals-13-02436-f002:**
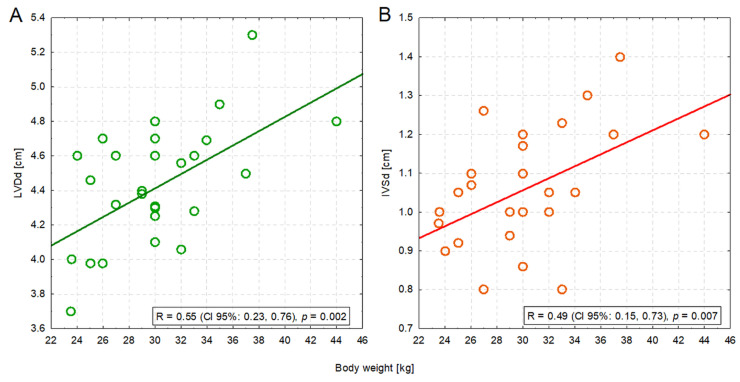
The moderate positive correlation between the body weight of dogs and the left ventricular dimension in diastole (LVDd) (**A**) or the interventricular septum in diastole (IVSd) (**B**).

## Data Availability

The data presented in this study are available on request from the corresponding author. The data are not publicly available due to the ethical agreement with the dogs owners that only anonymized, summary data would be publicly presented.

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
