# Peer review of "Cardiological Reference Intervals in Adult American Staffordshire Terrier Dogs"

_animals, 2023, doi:10.3390/ani13152436_

Round 1
Reviewer 1 Report
Comments to Author:
Manuscript ID: animals-2447561 entitled "Cardiological reference intervals in adult American Stafford-2 shire Terrier dogs".
This study attempted to determine reference intervals for echocardiography, ECG, vertebral heart score, and systolic blood pressure in American Staffordshire Terrier dogs. I have the following suggestions and questions.
1. The primary limitation of the study is the small number (n=29) of the study dogs, as the authors indicated in Line 304-305. Therefore, additional information should be provided to help readers understand how to effectively utilize these data.
(1) I understand that sometimes it is not feasible to have enough numbers in clinical animals. Could you please calculate what sample size is adequate for the study purpose? This information will provide an idea of the statistical power of the current study or the limitation in clinical reality.
(2) In the discussion: Since this manuscript pertains to normal reference values, it would be beneficial for the authors to include a brief literature review discussing the establishment of these reference values in small animal cardiology and the methodologies previously employed."
2. Other specific questions or suggestions:
(1) Line 27: “Compared to RIs available for the general population of dogs” & Line 205: “Compared to commonly accepted RIs……” Where is the data of “General population”?
(2) Line 49: Rephrase the sentence to improve clarity. (eg., The American Staffordshire Terrier is one of the canine breeds known for its distinctive body conformation)
(3) Line 76-78: “Only measurements obtained from dogs with normal results of clinical examination and blood check-up were used for determination of the RIs.” Please indicate the total number of dogs screened in the Results section, and specify that 29 of them were included in the final study population.
(4) Line 87-89: Please delete the cuff number such as no. 8 or 10 because it could be different by various manufacturers. Instead, please describe the width of the cuffs as indicated in the ACVIM 2019 Hypertension Guideline.
(5) Line 90-92: “SBP was measured 5 times, the lowest and highest values 90 were discarded, and the arithmetic mean of the middle 3 measurements was considered 91 the final SBP” This method might have been used in the past, but it is no longer acceptable following the release of the ACVIM 2009 and 2019 Hypertension Guideline. Since the current study has already been conducted, it is impossible to rectify this issue. Therefore, it should be acknowledged as a limitation in the discussion. Alternatively, the authors could express their concern regarding this modification in the materials and methods section.
(6) Line 101: Could you please explain why the QT interval was not adjusted by HR? What was the concern or rationale?
(7) Line 109: Replace “cardiac rhythm” with “ECG”.
(8) Line 112: Could you please state the reason for scanning at standing position?
(9) Line 121-126: Please specify the M-mode measurements were made from right parasternal short-axis papillary muscle view, or right parasternal short-axis chordae tendineae view?
(10) Line 132-134: Could you please explain why the 2D EF was measured from Rt para LX 4C and Lt apical 4C, rather than Lt apical 4C and 5C? A citation (or citations) should also be provided here.
(11) Line 154-161: The data associated with the statistical method listed here is unclear in the manuscript. Please provide more explanation or revise accordingly.
(12) Line 171-177 & Ref 21 & 22: The references here are for clinical pathology. Have the authors consulted with a statistician? Could you please clarify whether the methodology here is also applicable for echocardiographic and electrocardiographic data? Please list proper citations of this.
(13) Line 241: “i.e. ↓ 150 mmHg” I don’t understand. Please correct it to form a complete sentence.
(14) Line 241-242: “In 2 dogs, arterial pressure was beyond the RI (mean systolic arterial pressure 167 mmHg and 178 mmHg).” Did these 2 dogs have repeated SBP measurements in 2-4 weeks to definitely rule out systemic hypertension? For example, the authors did not perform urinalysis and ocular exam, thus TOD such as retinopathy and proteinuria was not really investigated.
(15) Line 253-254: “It should not be guided by one even very high measurement.“ Please rephrase to clarify the meaning.
The Quality of English Language: No serious problem.
Reviewer 2 Report
The paper highlights the need for breed-specific reference intervals for various cardiovascular parameters in American Staffordshire Terriers (ASTs). It suggests that by establishing these reference intervals, individual dogs can be assessed more accurately, increasing the likelihood of detecting even minor deviations from the breed-specific norms and preventing misdiagnosis. However, it seems challenging to provide a comprehensive evaluation of the study's quality and significance.
These results provide baseline data on clinically healthy dogs with asymptomatic systemic hypertension. The absence of cardiac abnormalities and the wide range of SBP values within the reference interval suggest a diverse but generally healthy population. Further research and comparisons with diseased populations would be valuable to understand the significance of these findings in the context of canine hypertension.
The adequacy of sample size for establishing statistical reference values depends on several factors, including the variability within the population, the desired level of precision, and the statistical methods used. While a sample size of 29 dogs may be sufficient for some analyses, it may not be adequate for establishing robust reference values in certain cases. To establish reference values with a high level of confidence, larger sample sizes are generally preferred. A larger sample size helps to minimize the effects of random variation and provides more precise estimates. Additionally, a larger sample size allows for better representation of the population and reduces the risk of bias. It is important to note that the number of dogs in a study should be determined based on statistical power calculations and the specific goals of the study. In some cases, smaller sample sizes may still provide valuable insights, especially in exploratory or preliminary studies. However, for more definitive and widely applicable reference values, larger sample sizes are typically recommended. In summary, while a sample size of 29 dogs may yield useful information, it may not be sufficient to establish highly accurate and precise statistical reference values. Larger sample sizes are generally preferred to ensure robustness and generalizability of the results.
The quality of the English is sufficient and that the content is understandable. Minor check for grammar is only required.
Round 2
Reviewer 1 Report
Manuscript ID: animals-2447561-peer-review-v2 entitled "Cardiological reference intervals in adult American Stafford-2 shire Terrier dogs".
I appreciate the revision made by the authors. They have diligently addressed all of my concerns, and I am grateful for their attention to detail. Their thoroughness in addressing the issues has significantly strengthened the quality of the work.